# Wnt/β-Catenin Signaling Pathway Governs a Full Program for Dopaminergic Neuron Survival, Neurorescue and Regeneration in the MPTP Mouse Model of Parkinson’s Disease

**DOI:** 10.3390/ijms19123743

**Published:** 2018-11-24

**Authors:** Bianca Marchetti

**Affiliations:** 1Department of Biomedical and Biotechnological Sciences (BIOMETEC), Pharmacology Section, Medical School, University of Catania, Via Santa Sofia 97, 95123 Catania, Italy; biancamarchetti@libero.it or bmarchet@unict.it; Tel.: +39-95478111194; +3355722698; 2OASI Research Institute-IRCCS, Section of Neuropharmacology, Via Conte Ruggero 73, 94018 Troina (EN), Italy

**Keywords:** Parkinson’s disease, cell death, dopaminergic neurons, Wnt/β-catenin signaling, glia–neuron cross-talk, neurodegeneration, neuroprotection, neurorepair

## Abstract

Wingless-type mouse mammary tumor virus (MMTV) integration site (Wnt) signaling is one of the most critical pathways in developing and adult tissues. In the brain, Wnt signaling contributes to different neurodevelopmental aspects ranging from differentiation to axonal extension, synapse formation, neurogenesis, and neuroprotection. Canonical Wnt signaling is mediated mainly by the multifunctional β-catenin protein which is a potent co-activator of transcription factors such as lymphoid enhancer factor (LEF) and T-cell factor (TCF). Accumulating evidence points to dysregulation of Wnt/β-catenin signaling in major neurodegenerative disorders. This review highlights a Wnt/β-catenin/glial connection in Parkinson’s disease (PD), the most common movement disorder characterized by the selective death of midbrain dopaminergic (mDAergic) neuronal cell bodies in the subtantia nigra pars compacta (SNpc) and gliosis. Major findings of the last decade document that Wnt/β-catenin signaling in partnership with glial cells is critically involved in each step and at every level in the regulation of nigrostriatal DAergic neuronal health, protection, and regeneration in the 1-methyl-4-phenyl-1,2,3,6-tetrahydropyridine (MPTP) mouse model of PD, focusing on Wnt/β-catenin signaling to boost a full neurorestorative program in PD.

## 1. Introduction

Parkinson’s disease (PD) is the most common movement disorder, and the second most common aging-related neurodegenerative disease after Alzheimer’s disease. The selective and progressive degeneration of midbrain dopaminergic (mDAergic) neurons of the substantia nigra pars compacta (SNpc) and their projections into the caudate–putamen (striatum) leading to substantial decrease in dopamine levels, the accumulation of pathological α-synuclein, which is a major component of Lewy bodies (LBs), and gliosis as the major hallmarks of PD [1,2]. As the disease advances, the progressive loss of mDAergic input to the striatum results in decreased motor function with symptoms that include bradykinesia, rigidity, tremor, and postural instability, although the underlying mechanisms leading to these symptoms are incompletely understood [1,2]. Additionally, non-motor symptoms, including cognitive impairment, as well as autonomic, olfactory, sleep, and mood disorders, frequently accompany the classic motor manifestations of the disease [1,2].

Both genetic mutations and exposure to environmental risk factors are linked to PD [3,4,5,6,7,8,9]. Indeed, several genes were identified in the rare familial (about 5%) forms of the disease, where autosomal dominant mutations in *SNCA*, *LRRK2*, and *VPS35*, and autosomal recessive mutations in *PINK1*, *DJ-1*, and *Parkin* cause the disease with high penetrance [9]. However, the majority of cases (almost 95%) are sporadic, underscoring a key role for gene–environmental interactions in PD [3,4,5,6,7,8,9]. In particular, aging, the leading risk factor for PD development, exposure to a panel of neurotoxicants, inflammation, and the hormonal background are key factors programming the vulnerability to PD [3,4,5,6,7,8,9,10,11,12,13,14,15,16,17,18,19,20,21,22,23]. The mechanisms underlying the selective and progressive mDAergic neuron death in PD and experimentally induced PD are not clarified; however, oxidative stress and inflammation associated to molecular changes indicative of mitochondrial dysfunction and apoptosis were identified in the parkinsonian brain [8,13,14,16,17,18,20,21,22,24,25,26,27].

So far, there is no cure for PD, and current treatments rely on dopamine replacement therapy, albeit only temporally alleviating the motor symptoms without stopping the ongoing neurodegeneration [1,2]. Therefore, an in-depth understanding of the molecular pathways chiefly involved in mDAergic neuron physiopathology is crucial for the development of neuroprotective and cell replacement therapies for PD.

Notably, the Wingless-type mouse mammary tumor virus (MMTV) integration site (Wnt)/β-catenin pathway is recognized as the vital player in mDAergic neurogenesis [28,29,30,31,32]. The evolutionarily conserved Wnt/β-catenin pathway initiates a signaling cascade that is crucial during both normal embryonic development and throughout the life of the organism in almost every tissue and organ system. Within the central nervous system (CNS), Wnt signaling cascades orchestrate all facets of neuronal functions, including differentiation, neuron death/survival, axonal extension, synapse formation and plasticity, neurotrophin transcription, neurogenesis, and regeneration [33,34,35,36,37].

Interaction of Wnt secretory glycoproteins with their seven-pass transmembrane receptors of the Frizzled (Fzd) family on responder cells initiate at least three different signaling pathways including the so-called “canonical” Wnt/β-catenin, and the “non-canonical” Wnt/planar cell polarity (PCP) and Wnt/Ca^2+^ pathways (Figure 1) [38,39,40,41,42,43].

A vast panel of endogenous ligands and receptors are involved in Wnt signal transduction. Furthermore, according to a specific cell tissue, Wnt/Fzd interactions and the target genes implicated may allow for different possible outcomes [40,41,42,43]. Such a complex role of Wnt signaling in virtually every organ system in normal homeostasis and repair after injury anticipates that its dysfunction may lead to an array of diseases, including neurological, inflammatory, and disorders of endocrine function [42,43].

Earlier and more recent findings underscored that Wnt1/β-catenin signaling cascades play a prominent role in mDAergic development [26,27,28,29,30]. Studies focusing on genetic networks recapitulating the early signals for the development of mDA neurons identified Wnt1 as a critical morphogen for mDA neurons, where activation of Wnt1/β-catenin signaling is required for mDA neuron specification [28,29,30,31,32]. In the adult brain, growing evidence highlights that a Wnt/Fzd/β-catenin tone is involved in the maintenance of neuronal health, while its dysregulation is associated with neuronal dysfunction and death [44,45,46,47,48].

This work traces the role of dysfunctional Wnt/β-catenin signaling in PD. Hence, the demonstration of the specific involvement of Wnt1 signaling components in the adult PD-injured midbrain associating with mDAergic neuronal cell death and self-repair in the 1-methyl-4-phenyl-1,2,3,6-tetrahydropyridine (MPTP) mouse model of basal ganglia injury uncovered, for the first time, a novel role for Wnt/β-catenin’s main actors in the maintenance and protection of mDA neurons in the adult and aged PD mesencephalon [49,50]. Interestingly, neuroinflammation appeared to play unsuspected roles, since Wnt1/β-catenin signaling and MPTP-reactive astrocyte cross-talk with both mDAergic neurons and microglial cells were unveiled as candidate components of the neurorescue pathways involved in nigrostriatal DAergic plasticity and in the regulation of neurogenesis [49,50,51,52,53,54,55,56,57], leading to the proposal of a novel “intrinsic Wnt/neuroinflammatory brain repair hypothesis” [51].

Importantly, environmental PD toxins and pesticides were shown to downregulate Wnt/β-catenin signaling in rodent, non-human primate, and human PD [58,59]. Accordingly, in genetic screens and in human post-mortem studies Wnt’s key components were shown to be altered [60,61]. More recently, other studies, looking at methylation sequencing, revealed dysregulated Wnt signaling in PD [62].

As the importance of Wnt signaling in midbrain DAergic neurogenesis was underlined by the use of chemical Wnt activators for efficient generation of mDA neurons from cultured pluripotent stem cells (PSCs) [63,64], an increasing number of studies focusing on Wnt/β-catenin signaling genes were reported. Notably, genome-wide analysis of gene expression identified Wnt signaling as an over-represented pathway in human induced pluripotent stem cell (hiPSC)-derived DA population [65], and activation of Wnt signaling was recognized to play an important role in deriving regionally homogeneous populations of neural stem cells (NSCs) and neurons [66], thereby greatly improving their scientific and therapeutic utility [66]. By contrast, pivotal PD proteins encoded by genes whose mutations were linked to PD were demonstrated to impact on canonical Wnt/β-catenin signaling [67,68,69] and to inhibit the ability of human induced pluripotent stem cells (iPSCs) to differentiate into DAergic neurons [70], whereas pharmacological Wnt activation restored their mDAergic developmental potential [70]. With the emerging role of micro-RNA (miRNA) regulatory functions on major pathways involved in mDAergic neurodegeneration in PD [71,72,73], patient-specific dysregulation of messenger RNA (mRNA)/miRNA expression is being revealed in PD [74], with an expected impact on Wnt signaling [43,75,76].

In addition to genetic mutations, environmental risk factors for developing PD, namely aging, environmental toxic exposure, and inflammation, were shown to impair and/or dysregulate Wnt/β-catenin signaling in mDAergic neurons and neural/stem progenitor cells (NSCs), thereby predisposing mDAergic neurons to apoptotic cell death and inhibiting the “intrinsic” mDA neurorepair potential with a severe impact on neurogenesis upon injury, while activation of Wnt/β-catenin signaling via either pharmacological or cell-specific therapies efficiently counteracted mDAergic toxity favoring neuroprotection and neurorestoration [49,50,51,52,53,54,55,56,57].

In particular, the sustained and ectopic expression of Wnt1 in genetically affected engrailed (En1) heterozygote (En1^+/−^) mice can induce a neuroprotective Wnt1-dependent gene cascade promoting the survival of En1 mutant (En1+/− and En1−/− mice) mDA neurons, rescuing them from premature cell death [77]. Remarkably, within the ventral midbrain (VM), astrocyte-derived Wnts decline with age, whereas the expression of endogenous antagonists of Wnt/β-catenin signaling, including Dickkopf1 (Dkk1), is upregulated, thereby contributing to the reduced neuronal survival and neurorepair capacity, and to the marked impairment of neurogenesis [49,50,51,52,53,54,55,56,57].

Of special importance, the inducible expression of Dkk1, resulting in deficient Wnt signaling, elicits synaptic degeneration in the adult striatum, associated to impaired motor coordination, which suggested that a dysfunction in Wnt signaling contributes to synaptic degeneration at early stages in neurodegenerative diseases [78]. Additionally, targeted deletion of β-catenin in DAergic neurons (DAergic-βcat knock-out (KO) mice) leads to alterations of motor and reward-associated memories and to adaptations of the DA neurotransmitter system [79]. By contrast, investigations in canonical Wnt reporter mice, BATGAL and Axin2LacZ, prooved the essential roles of Wnt/β-catenin signaling in mDAergic neurons and midbrain DAergic neuroprogenitors during degeneration, repair, and regeneration of nigrostriatal neurons in the aged PD brain [57,80]. The growing field of Wnt signaling in neurodegeneration and regeneration was highlighted in a Special Issue “Wnt signaling cascades in neurodevelopment, neurodegeneration, and regeneration”, featuring the progresses achieved and the future challenges in the field [81]. So far, a wide panel of genetic, physiopathological, or neurotoxic conditions affecting mDA neurons in PD were shown to strongly impair canonical Wnt/β-catenin signaling, while an increasing number of pharmacological and immunomodulatory agents affording neuroprotection were recognized to activate the canonical Wnt/β-catenin signaling pathway, promoting neuroprotection and immunomodulation, and counteracting the impairment of neurogenesis in PD injured brain (Figure 2) [49,50,51,52,53,54,55,56,57,82,83,84,85,86,87,88,89,90,91].

Based on this background, this work highlights Wnt/β-catenin signaling and its cross-talk with survival and neuroinflammatory pathways as a pivotal actor for mDAergic neuronal maintenance, protection, repair, and regeneration in mice affected by MPTP-induced nigrostriatal degeneration.

After a brief introduction of Wnt/β-catenin signaling, its potential role in dictating mDAergic neuron vulnerability to major key risk factors such as aging, gene mutations, neurotoxin exposure, and neuroinflammation is summarized, along with the potential therapeutical implications for PD in the light of recent published findings in the field.

## 2. The Key Interactors of the Wnt/β-catenin Signaling Cascade

In the central nervous system (CNS), the glycoprotein ligand Wnts play critical roles during development, and control neuronal integrity and homeostasis in adult brain through engagement of Fzd receptors [92]. Importantly, Wnts and the components of Wnt/β-catenin signaling are widely expressed in midbrain, and are required to control the fate specification of DAergic neurons [26,27,28,29,30,31,32].

Wnts (Wnt1–Wnt19) are ~350-residue secreted, cysteine-rich glycoproteins that locally activate receptor-mediated Wnt signaling [34,35,36,37,38,39,40,41,42,43]. Generally, three different signaling pathways are recognized to mediate the effects Wnts and include the “canonical” β-catenin pathway, and the “non-canonical” pathway, which comprises two arms: the planar cell polarity (PCP) and Ca^2+^ pathways (Figure 1). Based on the expression profile of Wnt ligands, the engaged Fzd (Fzd1–Fzd10) receptors, and co-receptors, coupled to the presence of endogenous cytoplasmic Wnt regulators in a given tissue, Wnt signals are then transduced to the canonical and non-canonical pathways [31,32,33,34,35,36,37,38,39,40,41,92,93,94].

Wnt proteins are usually referred to as the Wnt1 (including Wnt2, Wnt3, Wnt3a, Wnt8, and Wnt8a) and the Wnt5a (including Wnt4, Wnt5a, Wnt5b, Wnt6, Wnt7a, and Wnt11) classes. The Wnt1-like proteins signal into the cell via a β-catenin-dependent pathway, where activation of the transcriptional activity of the multifunctional protein β-catenin, represents the pivotal mediator of the canonical Wnt signaling. The Wnt5a class signals via β-catenin-independent “non-canonical” Wnt/PCP and Wnt/Ca^2+^ pathways. Of note, this functional Wnt classification is an oversimplification, as there are cases or contexts where the same Wnt protein activates different pathways depending on the presence of receptors) (Figure 1).

In addition to Fzd, the low-density lipoprotein receptor-related proteins 5 and 6 (LRP5/6) encountered on the cell surface are required for the activation of Wnt-dependent signaling. Another important component is Dishevelled (Dvl/Dsh), a cytoplasmic multifunctional phosphoprotein, acting at either the plasma membrane or in the cytoplasm in all three Wnt/Fzd signaling cascades. Notably, depending on the specific Wnt ligand and receptor context, Dvl can activate canonical Wnt signaling via the inhibition of β-catenin degradation, resulting in the stabilization of β-catenin and its nuclear translocation. Additionally, Dvl can regulate some of the non-canonical branches of Wnt signaling [92,93,94]. Alternatively, Fzd receptors can act independently of LRP5 or LRP6, function as receptors for Wnt, and activate β-catenin-independent pathways (Figure 1). Of mention, Wnt signaling is also regulated by several alternative receptors, such as the transmembrane receptor Tyr kinases Ror2 and Ryk members (Figure 1**)**.

Of specific interest, the leucine-rich repeat kinase 2, LRRK2, a member of the leucine-rich repeat kinase family, codified by *PARK8* gene, interacts with Dvl proteins [67]. LRRK2 is recruited to membranes following Wnt stimulation, where it binds to the Wnt co-receptor LRP6 in cellular models [67]. Pathogenic *LRRK2* mutations disrupted Wnt signaling, implicating binding to LRP6-mediated Wnt signaling caused by reduced binding to LRP6 as a potential factor underlying neurodegeneration observed in PD [67,68]. On the other hand, the protective LRRK2 R1398H variant enhanced GTPase and Wnt signaling activity [69], underlying the complexity of LRRK2/Wnt signaling cross-talk in PD [67,68,69]. Along this line, proteomic analysis of LRRK2 binding partners revealed interactions with multiple signaling components of the Wnt/PCP pathway [95].

In the canonical pathway (i.e., in the “Wnt on” state), Wnt1-like signals are transduced to the nucleus via β-catenin through the interaction of Wnt with Fzd receptor and the co-receptor LRP5/6, leading to the activation of Dvl. These events stabilize intracellular β-catenin through the Axin complex, composed of Axin, the tumor suppressor adenomatous polyposis coli gene product (APC), casein kinase 1 (CK1), and glycogen synthase kinase 3β (GSK-3β), resulting in the prevention of β-catenin phosphorylation and ubiquitin-mediated degradation (Figure 1A). Here, Wnt signaling-mediated inhibition of GSK-3β activity increases the amount of β-catenin, which enters the nucleus, and associates with T-cell factor/lymphoid enhancer binding factor (TCF/LEF) transcription factors, leading to the transcription of Wnt target genes involved in cell survival, proliferation, and differentiation. Notably, the endolysosomal compartment can positively regulate Wnt signaling by sequestering GSK-3β inside microvesicular bodies (MVBs), thus diminishing the cytosolic availability of the β-catenin destruction complex [96]. By contrast, defective endolysosomal compartment, leading to reduced sequestration of GSK-3β, may result in increased β-catenin degradation and downregulation of the canonical Wnt/β-catenin signaling, as observed in Gaucher’s disease (GD) iPSC neuronal progenitors, leading to marked alteration of NSC–DAergic differentiation potential (as summarized in Section 4) [70]. Notably, besides the Wnt/β-catenin pathway, GSK-3β is active in a number of key intracellular signaling pathways that impact on a multitude of cellular functions including cellular proliferation, migration, neurogenesis, inflammation, and apoptosis [97,98,99,100,101]. Of special mention for this subject, numerous apoptotic conditions leading to mDAergic neuron death antagonize Wnt/β-catenin signaling and can be facilitated by GSK-3β activation [100,101]. In the absence of Wnt (i.e., in the “Wnt off” state), cytoplasmic β-catenin is constantly degraded by the action of the Axin complex. CK1 and GSK-3β sequentially phosphorylate the amino terminal region of β-catenin, resulting in β-catenin ubiquitination and proteasomal degradation [93]. The consequent prevention of β-catenin from reaching the nucleus then represses Wnt target genes by the DNA-bound T-cell factor/lymphoid enhancer factor (TCF/LEF) family of proteins (Figure 1A).

The physiological role of canonical Wnt/β-catenin signaling is best exemplified during early brain development, when DAergic neurons originating from the ventral midline (VM) of the mesencephalon require Wnt1 as the prime mover of DAergic neurodevelopment, as it drives the initial origin for the midbrain DAergic progenitors, and consequently activates Wnt/β-catenin signaling to promote DAergic neurogenesis [28,29,30,31,32,102]. By contrast, removal of β-catenin in tyrosine hydroxylase-positive (TH^+^) neural progenitor cells in the ventral midbrain (VM) region negatively regulates mDAergic neurogenesis [28,29,30,31,32,102]. Here, β-catenin depletion interferes with the ability of committed progenitors to become DA neurons, resulting in adult animals with a significant loss of TH^+^ neurons in the adult VM [102]. Notably, excessive Wnt signaling is also detrimental for mDA neuron production, adding to the general notion that morphogen dosage must be tightly regulated [28,29,30,31,32,103].

In the β-catenin-independent Wnt/Ca^2+^ pathway, Wnts bind to Fzd resulting in stimulation of heterotrimeric G proteins, which further activates phospholipase C. This leads to increased Ca^2+^ release and activates two kinases: Ca^2+^/calmodulin-dependent protein kinase II (*CamK*II), and protein kinase C (PKC). This, in turn, stimulates transcription factors such as cyclic adenosine monophosphate (cAMP) response element-binding protein-1 (Figure 1B). In the PCP pathway, Wnt proteins bind to Fzd, which activates small GTP-binding proteins Rho and Rac and c-Jun N-terminal kinase via Dvl. This interaction results in cytoskeletal regulation and involves polarized cell shape changes and migration (Figure 1). Additionally, these downstream events may interact with each other and antagonize the β-catenin pathway at multiple points (Figure 1B).

Because Wnt/β-catenin is a powerful pathway, too much or too little might be detrimental and, thus, it must be tightly regulated. So far, various natural inhibitors/modulators of Wnt signaling pathway were identified which can antagonize or regulate Wnt signaling pathway [Figure 1]. Generally, depending on their functional mechanism, Wnt signaling inhibitors are divided into two classes: secreted Frizzled-related proteins (sFRPs) and Dickkopfs (Dkks) [104,105]. The members of the sFRP family, WIF (Wnt inhibitory factor)-1 and Cerberus, primarily bind to Wnt proteins and, thus, regulate the association of Wnt ligands to their transmembrane receptors, inhibiting both canonical and non-canonical signaling pathways (Figure 1). Members of the Dkk class bind to LRP5/6 component of the Wnt receptor complex to inhibit (Dkk-1, -2, -4) or activate (Dkk-3) canonical Wnt signaling [30], with one of the best characterized members being Dkk1. In addition to antagonists, Wnt signaling pathway is also activated and regulated by some secreted proteins acting as agonists, for example, R-spondin (Rspo) [38,39,40,41].

Importantly, while sFRPs are considered Wnt signaling antagonists, recent studies showed that specific family members can positively modulate Wnt signaling [104,105]. Additionally, Wnt/Fzd binding and cooperation with particular co-receptors, such as LDL receptor-related protein 5/6 (LRP5/6), receptor tyrosine kinase-like orphan receptor 1/2, or receptor-like tyrosine kinase, can define downstream signal specificity (Figure 1B). For a more comprehensive and historic perspective, we refer readers to earlier and more recent reviews [39,40,41,42,43,104,105].

During embryonic development, Wnts are expressed abundantly in close vicinity of DAergic neurons and regulate genetic networks that are required for the establishment of progenitor cells and terminal differentiation of mDAergic neurons in the later stage of embryogenesis [28,29,30,31,32]. In the adult brain, the maintenance of intracellular β-catenin levels is crucial for the regulation of neuronal homeostasis and synaptic plasticity, whereas depletion of β-catenin levels correlates with synaptic loss and mDAergic dysfunction [33,45,46,47,78,79]. Of specific note, given the complexity of the Wnt signal transduction network, a dysregulation through each of the Wnt pathways and/or the cross-talk between them leading to either hypo- or aberrant functioning may promote diverse pathogenic outcomes. Therefore, modulation of Wnt signaling is actively targeted for a number of physiopathological conditons including cancer, regenerative medicine, stem cell therapy, bone growth, and wound healing [34,35,37,42,43,76,104,106].

## 3. The MPTP Mouse Model Highlights Neuroinflammation as a Critical Risk Factor for PD

One of the most highly compelling pieces of evidence for the potential contribution of environmental toxic substances and neuroinflammation in PD was revealed in humans who developed a profound parkinsonian syndrome after intravenous use of street preparations of meperidine analogs which were contaminated with MPTP, a substance structurally similar to the herbicide paraquat [4,6,107,108]. Interestingly, in some patients MPTP-induced PD appeared almost immediately after exposure, whereas, in others, onset became evident only months or years later. MPTP was subsequently shown to injure, in a selective manner, the DAergic neurons in the nigrostriatal system [4,6,107,108,109,110]. Years later, post-mortem examination of persons exposed to MPTP showed ongoing DAergic cell loss without Lewy body formation and a marked microglial proliferation in the SNpc, with clusters of reactive microglia around nerve cells [4,6,107,108]. This finding was suggested to reflect an ongoing neurodegenerative process that persisted years after the initial toxic injury and that could have been perpetuated, at least in part, by chronic neuroinflammation [4,6,107,108]. Following this seminal discovery, MPTP was tested in various animal species, including non-human primate, showing its ability to recapitulate most, albeit not all, PD-like symptoms [109,110]. In monkey, MPTP administration was able to reproduce most of the clinical and pathological hallmarks of PD [110], and the degeneration of DAergic neurons in mice [108,109].

As recently reviewed by Dr. J.W. Langston [6], over the last 30 years, the identification of MPTP and its mechanism of action had a remarkable impact on PD physiopathology and the development of drug treatments for PD. So far, the MPTP mouse model is recognized as a valuable tool, and is widely used both in in vivo and in vitro cell models to understand the underlying molecular mechanisms of DAergic neurodegeneration in sporadic PD [6,109,110].

The discovery of the different metabolic steps playing essential roles in MPTP-induced neurotoxicity further represents a critical milestone in the “MPTP story” [6]. Given that MPTP is a highly lipophilic compound, it rapidly crosses the blood–brain barrier, and, after parenteral administration, levels of the toxin are already detectable in the brain within minutes. MPTP by itself is not a toxic substance; however, once in the brain, MPTP is metabolized to 1-methyl-4-phenyl-2,3- dihydropyridinium (MPDP) by the enzyme monoamine oxidase B (MAO-B) in non-dopaminergic cells (i.e., astrocytes) [4,6]. Next, MPDP is oxidized to the active MPTP metabolite, MPDP1/MPP^+^, which is then released into the extracellular space, where it is taken up by the dopamine transporter (DAT) and is concentrated within the nigral DAergic neurons where it inhibits complex I of the mitochondrial electron transport chain, resulting in ATP depletion and subsequent neuronal cell death [4,6]. In particular, the induction of oxidative stress results in the opening of mitochondrial permeability transition pore (mPTP), the release of cytochrome C, and the activation of caspases [25,101]. Of specific mention, mitochondrial damage due to Ca^2+^ overload-induced opening of mPTP is believed to play a key role in selective degeneration of nigrostriatal DAergic neurons in PD [24,25,101].

In non-human primates, and depending on the application paradigm used, MPTP can produce an irreversible and severe parkinsonian syndrome that replicates the majority of PD features, including rigidity, tremor, slowness of movement, and also freezing [110]. These effects are associated with severe mDAergic neuron loss (>90%) in the putamen and caudate nucleus, similar to the striatal DAergic decline in the brain of parkinsonian patients [110]. Indeed, a major effect of MPTP regards the loss of tyrosyne hydroxylase (TH)-positive neurons and TH-positive fibers in the striatum. Both age and the extent of toxicant exposure affect the severity of nigrostrial lesion and loss of DAergic functionality [109,110]. In mice, the systemic or intracerebral application of MPTP can also lead to severe damage of nigrostriatal DAergic neurons, including symptoms of motor control disturbances, ressembling those in human PD, such as akinesia, rigidity, tremor, gait, and posture abnormalities [110]. Regimen of MPTP administration in mice determines the mode of neuronal cell death in the subtantia nigra.

Of paramount importance was the discovery that glial inflammatory mechanisms were revealed to take place in the PD brain and were progressively recognized to contribute to nigrostriatal DAergic degeneration [4,6,7,11,12,13,14,15,16,17,18,19,20,21,22,47,48,49,50,51,52,53,54,55,107,108]. Then, in synergy with the early MPTP-driven mitochondrial impairment accounting for approximately 10% of DAergic neuronal death, astrocyte and microglial reactions were found to play dual “beneficial/harmful” roles in DAergic neurodegeneration, as highlighted in the “To be or not to be inflamed” article [14], and further documented in more recent reviews in the field [51,52,53,56].

In addition to MPTP, several environmental toxicants such as herbicides and pesticides, related to rural living/occupation in agriculture, were implicated as risk factors in PD, as epidemiologic studies found increased risk of PD associated with exposure to pesticides, solvents, metals, and other pollutants, and many of these compounds recapitulate PD pathology in animal models [3,4,5,6,7,8]. For all of these toxins, it is important to underline the interaction between the period of exposure and age at time of exposure. For example, the effect of systemic exposure to MPTP varies as a function of its schedule of administration (i.e., acute vs. chronic), and the age of the organism, with nigrostrial DAergic neurons of older individuals being more susceptible than DAergic neurons of young adult organisms [8,10,11,12,109,110]. On the other hand, toxic exposures that occur early in development (both single or combined exposures) could determine long-term pathology. Importantly, there are also hormonal influences such as gender and the estrogen background, as well as the stress hormones and glucocorticoid receptor levels, that contribute to the programming of mDAergic neuron vulnerability as part of gene–environment interactions, with a critical role played by glia and its proinflammatory mediators, via astrocyte–microglia–neuron cross-talk [8,12,14,17,18,19,23,111]. 

Notably, during the last three decades, different laboratories, including our own, underscored the pivotal role of astrocyte and microglial mediators in the parkinsonian brain, as documented in epidemiological, post-mortem, and animal studies, highlighting glia as a common final pathway directing toward neuroprotection/neurodegeneration [8,13,14,16,17,18,19,20,21,22,23,49,50,51,52,53,54,55,56,57,80,111]. As anticipated, aging represents the leading risk factor for the development of PD [8,10,11,12]. Hence, aging exacerbates inflammation and oxidative stress, which are the crucial hallmarks of MPTP, or 6-hydroxidopamine (6-OHDA)-induced PD, affecting plastic and regenerative responses [53,54,55,56]. As a consequence, young adult rodents experience time-dependent recovery/repair from MPTP insults, whereas aging mice fail to recover for their entire lifespan [56].

In fact, with advancing age, the compensatory potential or “adaptive capacity” of nigrostriatal DAergic neurons, believed to mask the ongoing mDA neurodegeneration in presymptomatic individuals, slowly diminishes, and the function of the nigrostriatal DAergic neurons is progressively impaired, leading to neurochemical, morphological, and behavioral changes [8,10,11,12,112,113,114,115]. With aging, gene–environment interactions further decrease the brain’s self-adaptive capacity, including the impairment of mDAergic compensatory mechanisms and the inhibition of neurogenesis, with harmful consequences for neuron–glia cross-talk, mDA neuron plasticity, and repair [49,50,51,52,53,54,55,56,57]. The critical involvement of Wnt signaling in gene–environment interactions in PD is summarized below.

## 4. Dysregulated Wnt/β-catenin Signaling Is Linked to Gene Defects in PD

Recent data from the study of genes linked to PD suggest a central importance of Wnt signaling pathways for the normal development and function of mDAergic neurons. The identification of at least 12 genes involved in familial PD, including *α-synuclein* (*SNCA*), *Parkin*, *Ubiquitin hydrolase*, *PTEN-induced putative kinase*, *DJ-1*, *leucine-rich-repeat kinase* (*LRRK2*), *vacuolar protein sorting 35 homolog gene* (*VPS35*), and *Glucocerebrosidase* (*GBA*), linked to autosomal dominant late-onset PD, provided new clues to the pathogenesis of PD [9,66,67,68,69,70]. Recently, whole-exome sequencing also described autosomal recessive *DNAJC6* mutations in PD cases [9]. However, the vast majority of PD cases are genetically complex, being the result of the combined action of common genetic variants in concert with environmental factors (see Section 3) [3,4,5,6,7,8,9]. Importantly, these gene products and their disease-associated mutations were shown to regulate several cellular pathways, including mitochondrial turnover, synaptic vesicle exocytosis/endocytosis, endosomal sorting, autophagy, and lysosomal functions that play vital roles in mDAergic neuron homeostasis [9,66,67,68,69,70,116,117,118,119,120,121,122,123,124,125]. In particular, Wnt signaling components appear to cross-talk with the majority of these critical pathways, thereby contributing to main cellular dysfunctions, as described. Hence, the studies of Dr. C.W. Berwik and K. Harvey with their collaborators [67,68,69] were the first to unveil an LRRK2/Wnt connection, as anticipated in Section 2. Notably, their recent studies clearly indicated that loss of LRRK2 causes increased canonical Wnt activity, both in vitro and in vivo [67,68]. Accordingly, over-expressed LRRK2 binds and represses β-catenin, suggesting LRRK2 may act as part of the β-catenin destruction complex [67,68,69]. Of special interest, some pathogenic LRRK2 mutations were demonstrated to enhance this effect while the protective R1398H variant relieves it [68,69]. Together, these data “strengthen the notion that decreased canonical Wnt activity is central to Parkinson’s disease pathogenesis” [68].

VPS35, codified by *PARK17* gene [117], is an essential subunit of the retromer complex. The VPS35 protein functions as a core subunit of a heteropentameric complex referred to as the retromer, and recent studies identified Wntless (Wls) and the retromer complex as essential components involved in Wnt signaling [118,119]. Emerging evidence indicates that the dominant mutation may act via both a toxic gain-of-function or a dominant-negative mechanism, or possibly via a potential combination of these mechanisms, and that VPS35 KO mice can also develop PD-like pathology [117].

Importantly enough, the earlier identification of Wntless, a sorting receptor for Wnt, and of the retromer, a protein complex required for the recycling of Wntless from endosomes to the trans-Golgi network [118,119], support a critical role for Wnt-dependent functions in “vesicle trafficking, recycling, and turnover, that may be central to the pathophysiology of PD” [118], with VPS35 mutations being potentially involved in dysregulation of Wnt secretion.

*GBA1*, a most frequent genetic risk factor for PD, encodes the lysosomal enzyme β-glucocerebrosidase (GCase) [70,120,121], with a reduced GCase activity resulting in accumulation of glucosylceramide and glucosylsphingosine in different tissues including the nervous system [70,120,121,122,123]. Recent reports showed that the endolysosomal compartment modulates canonical Wnt/β-catenin signaling ([124] and references therein). Of specific interest, using iPSCs derived from patients with *GBA1* mutation, Awad and colleagues [70] showed a dramatic decrease in the survival of DAergic progenitors due to the interference with Wnt/β-catenin signaling. Consistent with mutant *GBA1*-dysfunctional Wnt signaling, NSCs also exhibited reduced expression of hindbrain progenitor markers and an increased expression of forebrain progenitor markers, which highlights the requirement for normal GCase activity during early stages of neuronal development [70]. In this work, the authors, therefore, underscored the Wnt/β-catenin pathway as a potential therapeutic target for neuronopathic GD [70].

As a further level of Wnt impact on gene–environment interactions, most of the genes mutated in PD are expressed in immune cells [125], supporting a Wnt/neuroinflammatory connection in PD [51]. Reportedly, Wnt/β-catenin signaling has both positive and negative effects on neuroinflammatory pathways in the brain after neuronal damage induced by stroke, trauma, and PD [51]. Within this context, the genetic background and exposure to various neurotoxic or inflammatory challenges can promote a self-perpetuating cycle of microglial-mediated mDAergic neurotoxicity, whereby dysfunctional astrocyte–microglia cross-talk, including a dysregulation of Wnt signaling, contributes to increase mDAergic vulnerability [14,49,50,51,52,53,54,55,56,57,125,126,127,128,129]. Notably, such feedforward cycles of chronic activation of microglia and chronic damage of mDAergic neurons are likely to play a decisive role for the severity of nigrostriatal DAergic lesion and the overall detrimental effects of SNpc neurons and, consequently, their capacity for neurorepair [51,53,56,57]. Therefore, different scenarios are suspected to contribute to DAergic neuron degeneration in PD, and rely on the interaction between the specific genetic background and a combination of environmental risk factors engendering a cascade of neurotoxic effects, finally directing toward neuronal cell dysfunction and eventual death. In this context, astrocytes (AS) and microglia (M) are the key players, mediating the effects of both genetic and environmental influences, including PD mutations, aging, hormones, endotoxins, and neurotoxins, as a function of the host-specific repertoire of genetic and environmental risk factors [14,51,53,56]. Notably, their uncontrolled activation (i.e., under inflammatory/neurotoxic exposure and upon brain injury) favor a switch toward the so-called AS-1 and M-1 harmful glial phenotype, which may directly affect neurons by releasing various molecular mediators, such as pro-inflammatory cytokines, reactive oxygen species (ROS), and reactive nitrogen species (RNS), which in turn perpetuate/exacerbate glial activation, resulting in increased mDA neuron vulnerability and cell death (Figure 3). Within this context, the pivotal role of Wnt signaling in directing either neuroprotection/degeneration is further discussed in Section 5.

## 5. A Wnt1 Self-Protective Regulatory Loop Engaged by Astrocyte–Neuron Cross-talk Safeguards mDAergic Neurons

Because astrocytes play key roles in the maintenance of neuronal homoestasis, energy metabolism, and in particular the defense against oxidative stress, an impairment of astrocyte–neuron cross-talk as a function of aging and neurodegenerative conditions may contribute to disease progression and impair the ability of mDAergic neurons to repair upon injury, reviewed in References [17,18,19,53]. Astrocytes are known to secrete both inflammatory and anti-inflammatory factors, as well as anti-oxidant, neurotrophic, and survival factors, and play a critical role in modulating microglial activity. Importantly, glial fibrillary acidic protein (GFAP)-expressing astrocytes can contribute to cell genesis both as stem cells and as important cellular elements of the neurogenic microenvironment, with implications for self-recovery/neurorepair [53,56]. However, under severe neurodegenerative conditions, astrocytes’ neuroprotective properties may be downregulated with harmful consequences for astrocyte–microglial and glial–neuron interactions, leading to increased mDAergic neuron vulnerability and/or neuronal death [53,56].

### 5.1. In Vivo Studies Uncover Wnt Signaling Intermediacy as Chief Player in Glial Modulation

In 2011, a first series of in vivo results implicated Wnt/β-catenin signaling and MPTP-reactive astrocytes in situ as a novel candidate component of neurorescue/repair pathways in MPTP-induced nigrostriatal dopaminergic plasticity [49,50]. Here, spatiotemporal analyses carried out in both striatum and VM of young adult MPTP-injured mice (from 3 h to 49 days post lesion (dpt)) supported the early and profound nigrostriatal DAergic impairment and underscored a gradual and remarkable histopathological and functional DAergic recovery with time [49]. Hence, a significant mitigation of mDAergic neuronal cell loss, an increase in TH-positive fibers, and a recovery of dopamine and its metabolites in the striatum accompanied the reversal of MPTP-induced motor deficit in young mice [49], highlighting the compensatory potential the nigrostriatal DAergic neurons [114,115]. These effects were accompanied by dynamic changes in both GFAP^+^ astrocytes and ionized calcium-binding adaptor molecule 1 positive (IBA1^+)^ microglia, restricted to both SNpc and striatum, clearly supporting glial contribution to both DAergic degeneration and repair [49]. Interestingly enough, wide gene expression analysis of 92 mRNA identified a major and early upregulation of pro-inflammatory chemokines and the canonical prototypical Wnt member, Wnt1, during MPTP-induced DAergic degeneration and self-recovery [49].

This gene response appeared Wnt1-specific, since neither Wnt3a nor Wnt5a showed significant changes during the study. When spatio-temporal analyses within the injured VM were then correlated to gene expression and protein levels of the key Wnt1 signaling components, which included Fzd-1 receptor, β-catenin, and GSK-3β, canonical Wnt signaling emerged as a novel player in adult midbrain DAergic neurons [49]. Remarkably, MPTP mice showed an early and sharp downregulation of Fzd-1 receptor and β-catenin both at a transcriptional and protein levels, at 12 h and 1 dpt, thus within the temporal window of the active degeneration phase [49]. By contrast, increased GSK-3β protein levels were observed 12 h after MPTP discontinuation, paralleling mDAergic degeneration and the time-course of microglial activation [49]. After this time, Fzd-1 and β-catenin started recovering within the temporal window of Wnt1 upregulation, and exhibited a sharp upregulation by 14 dpt to reach control levels by 42 dpt. Importantly, between 3 and 14 dpt, a significant decline of active GSK-3β levels was measured, when ameboid IBA1+ microglial cells in striatum and SNpc were significantly reduced. These findings indicated that MPTP induced Fzd1/β-catenin downregulation and GSK-3β activation within the temporal window of active DAergic degeneration and microglial activation, whereas the recovery of canonical Wnt/β-catenin was associated with the gradual DAergic self-repair phase and downmodulation of microglial activation [49]. In situ hybridization histochemistry coupled to immunofluorescent staining next revealed MPTP-reactive astrocytes as a candidate source of Wnt1 in the injured PD brain, thereby suggesting astroglial-derived Wnt1 might provide a compensatory mechanism to limit the degenerative process and/or activate the spontaneous SNpc self-repair program [49].

In order to understand the relevance of Wnt/β-catenin signaling in the intact midbrain, Wnt1/Fzd interaction was antagonized by Dkk1 infusion within the left SNpc of young adult mice [50].

Here, Dkk1 caused a time-dependent decrease of TH^+^ neuron numbers in the ipsilateral infused, but not in contralateral uninfused SN, whereas unilateral infusion of saline within the SN did not change TH^+^ neuron number in either ipsilateral or contralateral SNpc. Interestingly, a marked increase in GFAP^+^ astrocytes and GFAP protein was observed within the ipsilateral Dkk1-infused SN 1–7 days post-infusion, as compared to contralateral uninfused SN or saline-infused SN, thereby indicating a time- and site-specific GFAP response to Dkk1 intranigral infusion, likely reflecting a potential compensatory response of reactive astrocyte to the acute interruption of Fzd/β-catenin signaling. Hence, dynamic changes of Wnt signaling components were correlated to the time-dependent loss of TH^+^ neurons, where downregulation of Fzd-1 receptor and β-catenin proteins was revealed in the ipsilateral as opposed to the contralateral SN. Of specific mention, β-catenin downregulation preceded and accompanied the TH^+^ neuron degeneration in the face of a marked upregulation of active GSK-β Tyr-216, the harmful player of canonical Wnt signaling. By contrast, simultaneous treatment of Dkk1-treated mice with a specific GSK-3β inhibitor efficiently counteracted mDAergic neuron death [49,50], thereby disclosing a paracrine canonical Wnt/β-catenin tone as an endogenous pathway linked to the survival/maintenance of adult midbrain DAergic neurons, while its interruption appeared associated to TH neurodegneration in SNpc [49,50].

### 5.2. In Vitro Studies Pinpoint Wnt1/Fzd1/β-catenin Survival Pathway for mDAergic Neurons

In vitro experiments further suggested activated VM astrocytes as likely components of the Wnt signaling pathway critically contributing to mDAergic neuroprotection against MPTP toxicity, as revealed in co-cultures of primary enriched mesencephalic DAergic neurons with VM primary astrocytes, when compared to neurons grown without astrocytes [49,50]. Hence, while in the monotypic neuronal cultures, the active MPTP metabolite, MPP^+^, induced the recognized DAergic toxicity as demonstrated by the dose-dependent decrease in the number of TH^+^ neurons paralleled by inhibition of tritiated dopamine, [^3^H]-DA incorporation; MPP^+^ application to astrocyte–neuron cultures resulted in a significant degree of neuroprotection [49,50]. In stark contrast, blocking Wnt/Fzd signaling with the prototypic inihibitor of canonical Wnt signaling, Dkk1, sharply antagonized VM astrocyte-induced DAergic neuroprotection [49,50]. Moreover, and of special interest, not only did VM astrocyte-derived Wnt signaling contribute to protect DAergic neurons, but it also promoted neurogenesis and DAergic neurogenesis from neural stem/progenitor cells (NSCs) derived from the adult midbrain (see Section 7) [49], thereby highlighting the unique role of astrocyte-derived Wnt1 signaling for DAergic survival, neurorepair and neurorestoration [49].

Next, the importance of an intrinsic Fzd-1/β-catenin tone was underscored using in vitro PD model systems and various cytotoxic conditions that mimic, both in vivo and in vitro, the biochemical characteristics of PD, namely oxidative stress and mitochondrial dysfunction [50]. Using real-time PCR, Western blotting, and immunocytochemistry, we first identified Fzd-1 receptor and β-catenin in purified mesencephalic neurons grown for 9–10 days in vitro (DIV), which supported the in vivo findings in the adult VM tissue [49]. The validity of this model was corroborated in parallel by application of MPP+, which confirmed the sharp loss of DA markers, in vitro, featuring a marked downregulation of Fzd-1 and β-catenin transcript and protein levels [50].

Hence, in three PD cellular models (i.e., serum deprivation, SD, 6-OHDA, and MPP^+^) applied to enriched primary mesencephalic neurons expressing the dopamine transporter (DAT) at high levels, a dramatic caspase3-dependent DAergic neuronal death followed the toxic challenge, whereas the preventive application of Wnt1 fully prevented DAergic neuron death and efficiently reversed β-catenin downregulation, both at mRNA and protein levels [50]. Further studies in both purified neurons and astrocyte–neuron co-culture paradigms, using pharmacological antagonism or RNA silencing along with functional studies, clearly established an intrinsic Wnt1/Fzd-1/β-catenin tone critically contributing to the survival and protection of DAergic neurons [49,50] (Figure 2). Hence, in β-catenin-depleted neuronal cultures via the introduction of β-catenin small interfering RNA (siRNA), the toxic effect of SD, 6-OHDA, or MPP^+^ was further amplified [50]. Additionally, upon β-catenin silencing, the ability of Wnt1 to protect TH+ neurons against 6-OHDA or MPP+ insult was abrogated, as revealed by the loss of TH^+^ neurons, neurite atrophy, and decreased [^3^H] dopamine incorporation, as compared to neuronal cultures treated with control siRNA, where Wnt1 treatment afforded full TH^+^ neuroprotection [50].

Looking at the response of the Fzd receptor family, it was found that DA neurons harbor most Fzd receptors, although Fzd-1 was almost 2–4-fold more abundant as compared to other Fzd members, showing either lower or undetectable levels. In addition, DA neuron exposure to either SD or 6-OHDA significantly downregulated only Fzd-1, while slight reductions or no changes were observed for all other Fzd transcripts, suggesting a specific Fzd-1 modulation in DAergic neurons in vitro upon the described challenges [50]. Furthermore, in the presence of Fzd-1 antisense oligonucleotides (Fzd-1AS), Wnt1 was unable to protect TH^+^ neurons from the different cytotoxic insults, as revealed by the failure to counteract the decreased TH+ neuron numbers and [^3^H] DA incorporation [50].

Another piece of evidence documenting the role of Wnt signaling in DAergic physiopathology was the correlation with a key component of canonical pathway, i.e., the multifunctional GSK-3β protein. As recalled in Section 2, in the absence of Wnt activity, increased active GSK-3β is known to phosphorylate β-catenin at serine or threonine residues of the N-terminal region, encouraging degradation of β-catenin through ubiquination [93]. Notably, GSK-3β is activated by phophorylation of Tyr-216 located in the kinase domain, and inactivated by phosphorylation of the N-terminal Ser-9. Importantly, during DAergic injury, the amount of phopshorylated GSK-3β Tyr-216 sharply increased, whereas the use of specific GSK-3β antagonists efficiently afforded GSK-3β Tyr-216 downregulation leading to DAergic neuroprotection [49,50]

Overall, these in vitro results indicated the ability of Wnt1 to activate β-catenin in mesencephalic DAergic neurons via Fzd-1 and the downregulation of active GSK-3β, thereby resulting in the stabilization of β-catenin and the transcriptional regulation of Wnt target genes. At the neuron–glial interface, astrocyte-derived Wnt1 via the Fzd-1 receptor may then transmit pro-survival signals to the impaired mDAergic neurons. Therefore, the blockade of GSK-3β-induced degradation of β-catenin, which in turn promotes neuron survival, represents a key step in Wnt1’s ability to safeguard mDAergic neurons [49,50]. Coupled to the in vivo studies, these findings corroborated Wnt/β-catenin signaling and its cross-talk with glial pathways as a novel actor in the neuron–glial scenario for the lifelong protection of the vulnerable mDAergic neuronal population (Figure 3).

## 6. Deficient Wnt Signaling with Age: A Leading Risk Factor for Nigrostriatal DAergic Neuron Integrity

As recalled in Section 3, one of the most critical risk factors for PD is represented by the aging process, associating with a gradual decline in the intrinsic capacity of DAergic neurons to recover, at least in part, via increased SNpc neuron vulnerability and dysfunctional glia–neuron cross-talk [8,9,10,11,12,21,54,55,56,57,113,114,115]. Given the critical role of the astroglial cell compartment during physiopathological aging [56], the relevance of canonical Wnt/β-catenin during the process of aging was next studied both in intact and MPTP-injured mice [49,50,51,52,53,54,55,56,57].

Hence, lack of Wnt1 expression in midbrain astrocytes in older as opposed to younger mice, and loss of β-catenin in mDA neurons correlating with the failure to recover in response to acute MPTP injury was first identified in 2011 [49]. Indeed, not only was the astrocyte-dependent neuroprotective response absent in aging mice midbrain, but major negative Wnt’s inhibitors, including GSK-3β, showed a significant upregulation and accompanied the failure of mDAergic neurons to repair upon MPTP injury [49].

Notably, the pharmacological activation of Wnt1/β-catenin signaling via downregulation of GSK-3β in old MPTP-injured mice mimicked nigrostriatal recovery documented in young mice, thus establishing a functional link between Wnt signaling and DAergic plasticity [49,50]. Hence, these findings raised the possibility that astrocyte-derived Wnt signals might directly and/or indirectly participate in DAergic neuroplasticity observed upon MPTP exposure of young adult mice. Accordingly, aging-induced counteraction of mDAergic neurorepair was mimicked by Wnt/Fzd signaling antagonism through intranigral infusion of Dkk1 in young mice (see Section 5). Likewise, increased Dkk1 expression was observed in the VM of aging mice [49,57], and in 6-OHDA-injured rodents [82].

Together, these results suggested that disruption of a key neurodevelopmental signaling pathway with age may predispose to loss of mDAergic plasticity via inhibition of Wnt/β-catenin signaling as a prelude for increased DAergic vulnerability and PD development [49,50]. Of specific interest, studies in the MPTP mouse model of PD [54,55] uncovered a synergy between aging, inflammation, and MPTP exposure within the subventricular zone (SVZ) niche, resulting in a marked downregulation of major components of the canonical Wnt pathway leading to a dysregulation of Wnt/inflammatory cross-talk and a dramatic impairment of neurogenesis [55]. Importantly, these events correlated with the failure of aged mice to exhibit the nigrostriatal DAergic recovery as observed in young mice [51,54,55]. In addition to the midbrain and the SVZ, a number of important studies [130,131,132,133,134,135] consistently implicated loss of Wnt signaling in the hippocampus as a key determinant for the impairment of neurogenesis during physiopathological aging [130,131,132,133,134,135].

Remarkably, aging and MPTP exposure further amplified Wnt/β-catenin down-modulation in SNpc tissues of 16–20-month-old mice, since the endogenous Wnt-antagonists, *Dkk1, sFRP2*, and GSK-3β showed a striking upregulation (of 6–14-fold) in face of exacerbated levels of inflammatory and oxidative stress genes, including *IL-1β*, *TNF-α*, *IL-6*, *iNOS2*, nuclear factor kappa B (*NF-κB*), and the phagocyte reduced nicotinamide adenine dinucleotide phosphate (NAPDH) oxidase, *Phox*, that accompanied the failure of nigrostriatal DAergic self-repair [57].

Of particular mention, the key neuroprotective role of Wnt1 in the aged midbrain was highlighted in the recent study of Zhang and colleagues in 2015 [77], in mice heterozygous for the homeodomain (HD) transcription factor (TF) Engrailed 1(En1^+/−^ mice), and characterized by the age-dependent and slowly progressing degeneration of the mDAergic neurons [77]. Here, the ectopic expression of Wnt1 in the adult En1^+^/Wnt1 VM activated a gene cascade that protected these genetically affected En1 heterozygote (En1^+/−^) neurons from their premature degeneration in the adult mouse VM. Hence, the direct Wnt1/β-catenin signaling targets lymphoid enhancer-binding factor 1 (*Lef1*), LIM homeobox transcription factor 1 (*Lmx1a*), fibroblast growth factor 20 (*Fgf20*), and *Dkk3*, as well as the indirect targets pituitary homeobox 3 (*Pitx3*; activated by *Lmx1a*) and brain-derived neurotrophic factor (*Bdnf*; activated by *Pitx3*), playing decisive roles in mDA neurodevelopment, were significantly upregulated [77]. The authors also showed that the secreted neurotrophin Bdnf and the secreted Wnt modulator Dkk3, but not the secreted growth factor Fgf20, increased the survival of En1 mutant DAergic neurons in vitro, suggesting that the “Wnt1-mediated signaling pathway and its downstream targets Bdnf and Dkk3 might, thus, provide a useful means to treat certain genetic and environmental (neurotoxic) forms of human PD” [77].

Furthermore, a major role of a physiological Wnt tone for synaptic maintenance and function in the adult striatum was recently uncovered [78]. The studies of Galli and colleagues first demonstrated the expression of several Wnts, their receptors, and modulators during synapse development and in the adult striatum [78]. The authors underscored a novel role for Wnt signaling in the maintenance/stability of excitatory and DAergic synapses in the adult striatum [78]. Then, in vivo blockade of Wnt signaling by inducibly expressing the secreted Wnt antagonist *Dkk1* resulted in a significant degeneration of DAergic cortico-striatal excitatory synapses in striatum and a decrease in glutamate release from cortico-striatal afferents [78]. Moreover, transgenic mice that overexpress *Dkk1* in the hippocampus exhibit synapse loss, impaired long-term potentiation, enhanced long-term depression, and learning and memory alterations [47,136]. Notably, the effect of *Dkk1* can be reverted by Wnt agonists or by the inhibition of GSK-3β, thereby indicating that “the endogenous Wnt proteins are critical to maintain synaptic connections in the adult brain” [137].

Taken together, these results suggest that, in PD brain, an early downregulation of Wnt signaling starting by middle age might predispose the vulnerable nigrostriatal DAergic neurons to dysfunction and/or death. In synergy with the aging process, different risk factors, including inflammation and MPTP exposure, may act in concert to impair Wnt signaling both inside and ouside the neurogenic niches, with harmful effects for nigrostriatal functionality and neurogenesis. Importantly, the potential exists to revert some of these age-dependent changes, which is discussed in Section 7.

## 7. Wnt/β-catenin Signaling Is Required for DAergic Neurorepair and Regeneration

In the adult brain, neural progenitors in neurogenic areas such as the SVZ and the subgranular zone (SGZ) of the hippocampus are in intimate contact with astrocytes which helps generate an instructive “niche” that promotes neurogenesis [138,139,140]. Notably, astrocyte-derived Wnts and Wnt/β-catenin signaling activation contribute to the regulation of adult neurogenesis [141]. During development, Wnt1/β-catenin activation controls DAergic neurogenesis by maintaining the integrity of the neurogenic niche and promoting the progression from nuclear receptor related 1 protein positive (Nurr1+)/TH− post-mitotic DAergic neuroprogenitors to Nurr1+/TH+ neurons [142,143]. Recently, the adult midbrain aqueduct periventricular regions (Aq-PVRs) were shown to harbor neural stem/progenitor cells (mNSCs) with DAergic potential in vitro [144]. However, restrictive mechanisms in vivo are believed to limit their DAergic regenerative capacity [144,145].

Interestingly, in vivo studies in young adult mice revealed that MPTP-induced SNpc neuron death promoted a remarkable astrocyte-dependent remodeling within the Aq-PVR niche [80]. Fascinatingly, these cells had the morphology of radial glia, the stem cell population in the CNS that persists in the adult brain. Hence, high levels of expression of Wnt/β-catenin genes, together with their Wnt-dependent transcription factor, Nurr1, were associated with a remarkable time-dependent nigrostriatal DAergic histopathological and functional neurorestoration upon MPTP injury [80]. These results are in line with earlier and more recent evidence indicating that Wnt/β-catenin signaling is required for radial glial neurogenesis and neuron regeneration following injury [146,147,148]. Hence, using Cre-mediated lineage tracing to label the progeny of radial glia, Wnt/β-catenin activation was necessary for progenitors to differentiate into neurons [147]. Furthermore, axonal regrowth following injury also required Wnt/β-catenin signaling, suggesting coordinated roles for this pathway in functional recovery [141,142].

Canonical Wnt/β-catenin signaling reporter mice are strains of transgenic mice with a LacZ transgene controlled by TCF/LEF consensus DNA-binding elements and a minimal promoter [149]. The establishment of transgenic Wnt reporter mice and reliable antibodies allows the identification of cell types that contain functional Wnt signaling, express Fzd receptors, and secrete Wnt ligands. Using transgenic (BATGAL and Axin) β-catenin reporter mice, Wnt/β-catenin signaling activation was next demonstrated both on Aq-PVR-DA niche and mDAergic neurons in response to MPTP [80]. Hence, in young adult mice, spatio-temporal analyses further unveiled β-catenin signaling in predopaminergic (Nurr1+/TH−) and imperiled or rescuing DAT+ neurons during MPTP-induced DA neuron injury and self-repair [80]. Currently, the neurogenic potential of DA neurons in the adult midbrain is a highly-debated issue [80,144,145,150,151,152,153]. Recent findings showed newborn DAergic neurons mainly derived from the migration and differentiation of the NSCs in the Aq-PVRs and their adjacent regions upon 6-OHDA lesion [153], thus supporting the possibility of new DAergic neuron formation in response to SNpc DAergic neuron death, as previously indicated [57,80].

As, with the aging process, Wnt signaling in the brain declines, this results in the impairment of Wnt-mediated self-protective, neuroreparative, and neurorestorative DAergic mechanisms. Of special interest, in the aged PD mouse model, the changing properties of the midbrain Aq microenvironment resulted in reduced DAergic neurogenic potential of Aq-NSCs via a loss of astrocytic Wnt1 and a failure of Wnt/β-catenin signaling activation [80]. This effect, in turn, was associated with the impairment of nigrostriatal DAergic recovery from MPTP insult of aged mice for their entire lifespan [80].

Ex vivo and in vitro studies coupled to different co-coculture paradigms and a panel of experimental conditions next indicated that both glial age and a decline of glial-derived factors, including Wnt1, were responsible for impaired NSC neurogenic potential within the SVZ [54,55] and Aq-PVR niches [80]. Notably, aged NSCs still retained their neurogenic and DA differentiation potential when Wnt/β-catenin signaling was restored via “astrocyte rejuvenation”-induced *Wnt1* expression or under Wnt/β-catenin activation regimens, such as GSK-3β antagonism, leading to DA neuron formation, in vitro [80]. In vivo studies further confirmed that the pharmacological activation of β-catenin in situ with a specific GSK-3β antagonist promoted a significant degree of DAergic neurorestoration associated with reversal of motor deficit, with implications for neurorestorative approaches in PD [80].

Finally, the unique role of astroglial-derived Wnt1 in aged MPTP-injured mice was further highlighted very recently, following NSC transplantation in the injured SNpc, which promoted a remarkable time-dependent nigrostriatal DAergic neurorescue/repair [57]. Here, SVZ-derived adult NSCs transplanted in the aged MPTP-injured SNpc mainly differentiated into astrocytes re-expressing *Wnt1*. In particular, these NSC-derived astrocytes promoted a remarkable time-dependent endogenous nigrostriatal DAergic neurorepair [57]. Astrocyte-derived factors, especially Wnt1, were shown to act at different levels to rejuvenate the host microenvironment and to promote DAergic neurorestoration in the aged MPTP-injured brain [57].

Taken together, these results suggest the potential to restore mDAergic neuron functionality by activating Wnt/β-catenin signaling via both pharmacological and cellular approaches aimed at rescuing the imperiled mDAergic neurons, downmodulating the exacerbated microglial reaction, and via the activation and/or the recruitment of endogenous mDAergic progenitors [57,80].

## 8. Targeting Wnt Signaling as a *“Wn(t)dow”* of Opportunity for mDAergic Neurorescue

In the last few years, an increasing number of pharmacological and immunomodulatory agents affording neuroprotection were recognized as activators of the canonical Wnt/β-catenin signaling pathway. Different studies focused on the neuroprotective capacity of Wnt1-agonists and pharmacological inhibitors of GSK-3β. Hence, in 2013, Wei et al. [83] supported the ability of exogenous Wnt1-induced activation of Wnt/β-catenin pathway, to protect SH-SY5Y cells against 6-OHDA-induced DA toxicity, and, in 2015, Zhang and colleagues [85] corroborated the protective role of enhancing β-catenin activity via GSK-3β inhibition to afford neuroprotection of PC12 cells against rotenone toxicity. Here, GSK-3β inhibitors LiCl and SB216763 leading to β-catenin stabilization afforded neuroprotection via the induction of the mDAergic transcription factor, orphan nuclear receptor, Nurr1 [85], crucially involved in the survival and maintenance of mDAergic neurons. Amongst others GSK-3β inhibitors, bromoinduru-30-oxime (6-BIO) was shown to protect hippocampal neurons from the apoptotic effects of amyloid-β (Aβ) oligomers via direct activation of the Wnt/β-catenin pathway [154]. Interestingly enough, different classes of pharmacological agents including statins (simvastin) [87], opioids (pentazocine) [88], nicotinic receptor modulators [89], or derivatives of natural products and immunomodulators [84,86,90,91,155,156], amongst others, were reported to protect mDAergic neurons against apoptosis, in either in vivo or in vitro models of PD, via the activation of the Wnt/β-catenin signaling pathway, thus supporting the critical role of this signaling system for the protection of mDAergic neurons against cytotoxicity.

Other studies indicated the potential of Wnt1-like agonists, such as Wnt1 inducible signaling pathway protein 1 (WISP1), a downstream target in the Wnt1 pathway, to block neurodegeneration [157,158,159]. WISP1, also known as CCN4, is a member of the six secreted extracellular matrix-associated CCN family of proteins that mediates a wide panel of critical functions including the ability to prevent apoptosis, control caspase activation, and oversee autophagy [157,158,159]. The neuroprotective mechanism of WISP1 was shown to involve pivotal pathways controlling neuronal death/survival, such as phosphoinositide 3-kinase/protein kinase B (Akt1) apoptotic mitochondrial signaling, and included B-cell lymphoma (Bcl-2)-associated death promoter (Bad), Bcl-2-associated X protein (Bax), Bcl-2-like protein (Bim), and Bcl-xL [157,158,159]. Thus, targeting downstream pathways of Wnt1, such as WISP1, may indicate potential avenues for neurorepair upon CNS injury.

The interrelationships among inflammatory, survival, and Wnt/β-catenin signaling cascades also uncovered a complete regulatory loop impacting in both neurogenesis and neuronal outcome upon injury, protecting mDAergic neurons from loss in the MPTP mice model of PD through inflammatory inhibition, via activation of PI3K/phosphorylated (p)-Akt and Wnt1/Fzd1/β-catenin cell signaling pathways [49,50,51,52,53,54,55,56,57,160,161], thus prompting further investigations along these lines.

Together, these and other findings of the last few years support the indication of Wnt/β-catenin signaling as a critical final common pathway for mDAergic neurorescue, with an increasing interest in its therapeutic targeting [37,42,43,106]. Given the implication of Wnt/β-catenin signaling in the control of tissue homeostasis during development and disease, unraveling its complex biological roles, especially in the aged PD brain and under different gene–environmental conditions, will further increase our knowledge on PD physiopathology and identify novel therapeutic avenues.

## 9. Concluding Remarks and Future Directions

A central challenge in the field of neurodegenerative disorders is developing therapeutic strategies that boost neuroprotection and neurorepair, and that help endogenous regenerative programs in the injured brain. This work summarized several lines of evidences supporting a key role for the canonical Wnt/Fzd-1/β-catenin pathway contributing to the maintainance of mDAergic neuron survival and functionality within the adult intact and PD midbrain. Accumulating data from the studies of genes mutated in PD underscore the pivotal role of Wnt signaling pathways for the function of mDAergic neurons in health and disease states. New potential associations between the Wnt pathway and mitochondrial dynamics, lysosomal biogenesis, vesicle trafficking, apoptosis, autophagy, immunoregulation, and cell cycle, affecting key functions in neuroprogenitors, post-mitotic neurons, and glial cells are starting to be unveiled.

Notably, Wnt1-induced neuroprotection is closely related to the astroglial response to oxidative stress and inflammation upon injury, and requires Fzd-1 receptor and β-catenin stabilization coupled to GSK-3β inhibition to promote mDAergic neuron survival. Reactive astrocytes upregulate Wnt/β-catenin signaling. This astrocyte-dependent property might help in mitigating excessive inflammation as observed during aging and neurodegeneration. Importantly, Wnt signaling at the neuroimmune interface plays a pivotal role in the regulation of neuroprogenitors, post-mitotic neurons, and microglial cell functions in PD. Also, pharmacological manipulation of microglial oxidative and nitrosative status, either in vitro or in vivo, can activate a beneficial Wnt/β-catenin signaling, thus affording neuroprotection and inducing a successful neurogenesis rescue. Therefore, modulating these astroglia changes may represent a powerful ground for novel therapeutic intervention strategies to prevent, delay progression, and/or ameliorate pathology.

Together, these studies suggest that Wnt-induced pathways are required for the regulation of a panel of processes linked to mDAergic neuroprotection, neurorepair, and regeneration after injury, and include the re-expression of genetic neurodevelopmental programs for DA neuroprogenitor acquisition of the mature mDAergic phenotype, DA neuron formation, and migration, as well as their survival and maintenance within a hostile microenvironment thanks to Wnt’s immunomodulatory properties.

Given that the activity of Wnt signaling in the brain declines with the aging process, this results in the inhibition of Wnt-mediated self-protective, neuroreparative/neurorestorative, and immunomodulatory mDAergic programs. Because Wnt/β-catenin signaling controls the expression of a wide panel of direct and indirect target genes, mis-regulation of this signaling cascade may be involved in various age-dependent diseases associated with impaired neurogenesis, including PD. Notably, the signaling mechanisms contributing to neuronal death and NSC impairment in the aged PD brain target the Wnt/β-catenin signaling pathway. Coupled to the increasing evidence on the key role of Wnt/β-catenin signaling cascades in neurodevelopment, neurodegeneration, and regeneration, developing targeted in situ pharmacological interventions/cell manipulations that boost the inherent Wnt-dependent regenerative potential have implications for both restorative and regenerative approaches in PD.

Notably, given the multitude of roles of Wnt/β-catenin signaling in the control of tissue homeostasis and disease, there is an increasing interest in targeting this pathway. From a therapeutic viewpoint, the Wnt signaling pathways presents several challenges to the development of a targeted neuroprotective drug, and further studies are still needed to elucidate the complex Wnt signaling circuitry. For example, the Wnt pathway itself is protective in different tissues, but the receptor context may influence the regulation of canonical Wnt signaling by agonist or antagonists. In addition, the cellular context, the presence of gene mutations, and the interaction with a panel of environmental risk factors, such as aging and inflammation, may finally mediate the tissue-/cell-specific response.

As a whole, the understanding of the intricate signaling networks of Wnt/β-catenin signaling is critical for the identification of new potential therapeutic targets and the development of pharmacologic and cellular approaches for neurodegenerative diseases, including PD.

## Figures and Tables

**Figure 1 ijms-19-03743-f001:**
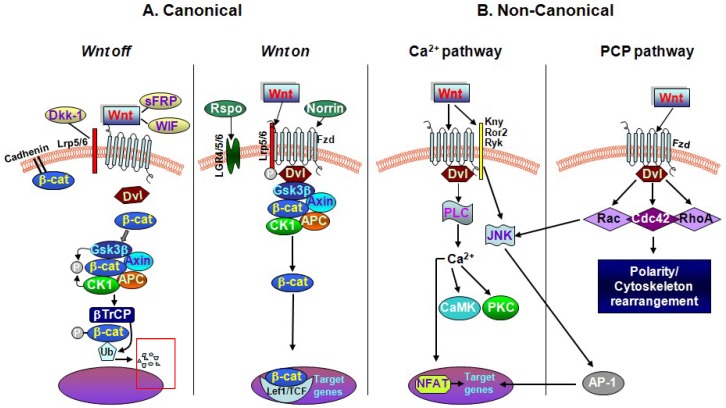
Schematic representation of canonical Wingless-type mouse mammary tumor virus (MMTV) integration site (Wnt) pathway and non-canonical Wnt/Ca^2+^ and Wnt/planar cell polarity (PCP) pathways [38,39,40,41,42,43]. (**A**) In the Wnt off state (i.e., in the absence of a Wnt ligand), cytoplasmic β-catenin is constantly degraded by the action of the Axin complex. Casein kinase 1 (CK1) and glycogen synthase kinase 3β (GSK-3β) sequentially phosphorylate the amino terminal region of β-catenin, resulting in β-catenin ubiquitination and proteasomal degradation (red box). In the Wnt on state, canonical Wnt/β-catenin pathway activation starts with Wnt binding Frizzled (Fzd) receptor and the co-receptor low-density lipoprotein receptor-related protein 5/6 (LRP5/6), which induces the recruitment of Dishevelled (Dvl) leading to the inhibition of the β-catenin destruction complex formed by Axin, adenomatous polyposis coli (APC), GSK-3β, and CK1. This inhibition causes the accumulation of β-catenin, which is no longer phosphorylated by the destruction complex. β-catenin then translocates to the nucleus where it activates transcription of Wnt target genes. (**B**) In the non-canonical Wnt/Ca^2+^ pathway, the binding of Wnt to Fzd activates the heterotrimeric G-proteins. These signal through phospholipase C (PLC) and inositol 1,4,5-triphosphate (IP3) to induce the release of intracellular Ca^2+^ and the activation of protein kinase C (PKC) and Ca^2+^/calmodulin-dependent protein kinase II (CaMKII). Multiple interactions are possible between the targets of the Wnt/Ca^2+^ and the canonical Wnt/β-catenin pathway at different points. Additionally, Kny, Ror2, or Ryk receptors with Fzd receptors can activate c-Jun N-terminal kinase (JNK), promoting target gene expression through activator protein 1 (AP-1). In non-canonical Wnt/PCP pathway, Wnt proteins bind to Fzd, which activates small guanosine triphosphatases (GTPases) proteins Rho and Rac and c-Jun N-terminal kinase via Dvl. This interaction results in cytoskeletal regulation and involves polarized cell shape changes and cytoskeleton rearrangement.

**Figure 2 ijms-19-03743-f002:**
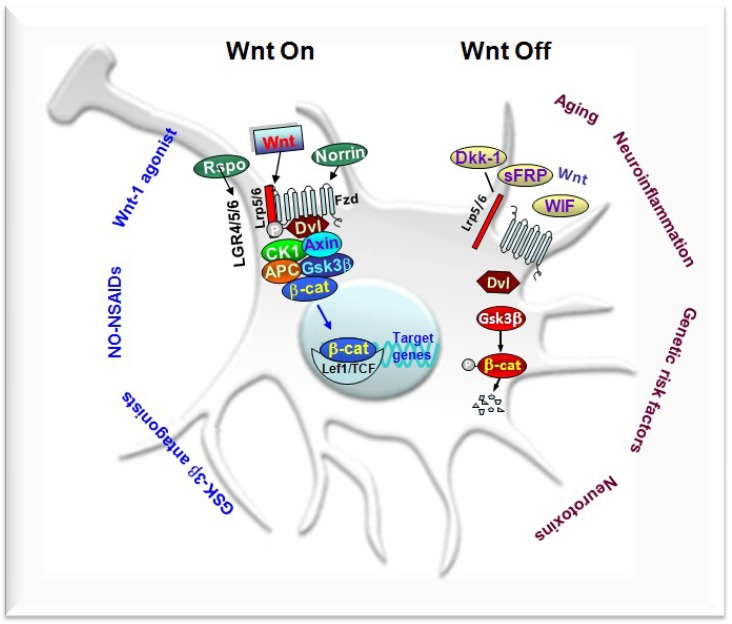
Schematic illustration of Wnt1/β-catenin signaling regulation of midbrain dopaminergic (mDAergic) neuron survival/death according to our published findings [49,50,51,52,53,54,55,56,57]. Major environmental factors including aging, inflammation, and 1-methyl-4-phenyl-1,2,3,6-tetrahydropyridine (MPTP) exposure may increase oxidative load and a panel of endogeneous Wnt antagonists (Dickkopf1 (Dkk1), Wnt inhibitory factor (WIF), frizzled-related proteins (sFRPs)) that, in synergy with genetic mutations and dysfunctional glia–neuron interactions, antagonize canonical Wnt/β-catenin signalling (“Wnt off”) in mDA neurons. Upregulation of active GSK-3β then leads to β-catenin degradation promoting DA neuron death. In the presence of canonical Wnt1-like agonists (such as R-spondin (Rspo) or Norrin), GSK-3β antagonists, or nitric oxide non-steroidal anti-inflammatory drug (NO-NSAID) treatments, activation of Wnt/β-catenin (“Wnt on”) contributes to maintaining the integrity of mDA neurons via blockade of GSK-3β-induced phosphorylation and proteasomal degradation of the neuronal pool of β-catenin. β-catenin then translocates into the nucleus and associates with a family of transcription factors regulating the expression of Wnt target genes involved in mDAergic neuron survival. Schematic drawing based on published results on Wnt signaling in PD [49,50,51,52,53,54,55,56,57].

**Figure 3 ijms-19-03743-f003:**
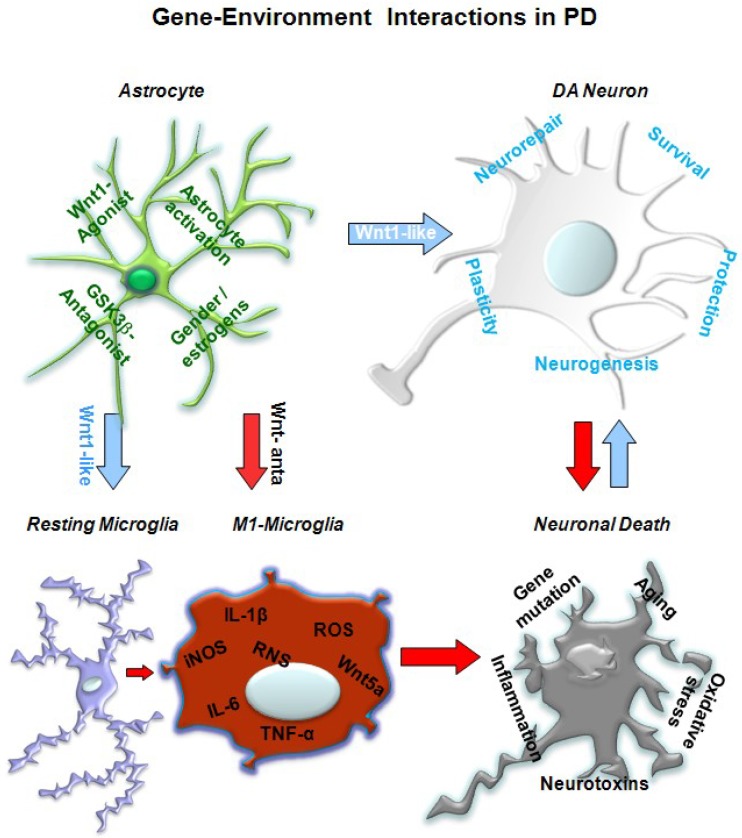
Schematic illustration of Wnt1/β-catenin signaling as a key player in glia–neuron cross-talk. A simplified scheme linking reactive astrocytes, mcroglia, and Wnt/β-catenin signaling to mDAergic neuron survival/death is summarized according to our published findings [49,50,51,52,53,54,55,56,57]. Upon injury, a number of conditons, including astrocyte activation, the genetic and hormonal background (i.e., gender and estrogens), and endogenous and exogenous activators of Wnt/β-catenin signaling components (i.e., GSK-3β-antagonists), can promote astrocyte beneficial effects via the expression of a panel of growth/neurotrophic factors, particularly Wnt1, contributing to the survival, repair, and neurorescue of DA neurons, via direct neuronal effects (see Figure 2) and through the inhibition of the microglia-M1 activated phenotype. Astrocytes of the ventral midbrain, via activation of Wnt/β-catenin signaling, can also promote neurogenesis and DAergic neurogenesis from adult neural stem/progenitor cells. By contrast, aging, MPTP exposure, and genetic mutations exacerbate microglia activation, with upregulation of a wide panel of pro-inflammatory mediators including tumor necrosis factor α (TNF-α), interleukin 1β (IL-1β), Wnt5a, inducible nitric oxide synthase (iNOS-derived) nitric oxide (NO) and reactive oxygen (ROS) and reactive nitrogen (RNS) species. Neurotoxic injury or increased oxidative load as a result of gene–environment interactions may antagonize Wnt/β-catenin signaling in DA neurons by upregulating active GSK-3β, leading to β-catenin degradation and increased DA neuron vulnerability.

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
