# Peer review of "Wnt/β-Catenin Signaling Pathway Governs a Full Program for Dopaminergic Neuron Survival, Neurorescue and Regeneration in the MPTP Mouse Model of Parkinson’s Disease"

_ijms, 2018, doi:10.3390/ijms19123743_

Round 1

Reviewer 1 Report

This MS is focused on a Wnt/beta catenin signaling in PD and recent studies in this field are well summarized. I believe that this review article will give a benefit to understand recent progress of Wnt/B-catenin signaling with glial connection in the brain disease. However, I found there are several sentence or paragraphs throughout the entire MS including figure legends, which should not be relived form the issue of plagiarism. Thus I recommend the author should carefully modify the sentences as his/her own voice. Below are the representative articles those I found the author might inappropriately quote.

"Uncovering novel actors in astrocyte-neuron crosstalk in Parkinson's disease: the Wnt/β-catenin signaling cascade as the common final pathway for neuroprotection and self-repair", European Journal of Neuroscience, 2013.

"The emerging role of Wnt signaling dysregulation in the understanding and modification of ageassociated diseases", Ageing Research Reviews, 2017

"Wnt/β-catenin signaling is required to rescue midbrain dopaminergic progenitors and promote neurorepair in ageing mouse model of Parkinson's disease : Wnt/β-catenin signaling drives DA plasticity", Stem Cells, 2014.

"Microglia Polarization, Gene-Environment Interactions and Wnt/β-Catenin Signaling: Emerging Roles of Glia-Neuron and Glia-Stem/Neuroprogenitor Crosstalk for Dopaminergic Neurorestoration in Aged Parkinsonian Brain", Frontiers in Aging Neuroscience, 2018

"Neural Stem Cell Grafts Promote AstrogliaDriven Neurorestoration in the Aged Parkinsonian Brain via Wnt/β-Catenin Signaling", STEM CELLS, 2018

"Wnt your brain be inflamed? Yes, it Wnt!", Trends in Molecular Medicine, 2013

Author Response

I Thank very much the reviewer for his/her important comments to this work.

As suggested, the entire manuscript has been reworded, sentences and paragraphs have been changed throught all the work, as well as the Figure legends. In addition, some parts of the work have been expanded according to Reviewers 2 and 3, while some sentences/paragraphs not focused on Wnt signaling in PD were not included.

Concerning the “plagiarism”, I would like just to comment : the mentioned reviews/articles (excepted for Ageing Res Reviews) belong to the bibliographic records of the B. Marchetti’s group of coworkers that over the years contributed to the results achieved in the field of inflammation, aging ,Wnt signaling and neurogenesis  in PD. All these papers as well as the original articles have been correctly cited in the Review, and it is quite “physiological” that some sentence or paragraphs might be similar (given that I wrote these articles and Review articles!). On the other hand, going through the Wnt literature, I have to remark, that a number of review papers/articles on Wnt signaling have entire paragraphs pasted from our papers, in other cases,  the authors omitt to cite our own work that first uncovered in 2011  Wnt’s key components in both adult and aging intact and PD brain, their role as key players during degeneration and repair/neurorestoration of nigrostriatal DA neurons, the decline of Wnt signaling with age, as well as Wnt crosstalk and relationships with major PD culprits such as oxidative stress and inflammation, aging, neuron-glial interactions, and the key role of the  Wnt/inflammatory connection for the regulation of adult neurogenesis in both intact adult and aged PD brain, together with the pharmacological implications for novel therapeutic strategies in PD.

Reviewer 2 Report

The author has well summarized the literature up to date of the Wnt/βcatenin signaling pathway in the regulation of dopaminergic neuronal survival, protection and regeneration in the MPTP mouse model of PD.

Overall, the review is well written and potentially interesting both for a specific and broad audience. However, the entire manuscript needs further work and improvement. The text, sometimes, is not that easy to follow and does not catch the full attention of the reader.

Major points

1. The quality/resolution of all the figures needs to be increased, adjusting also the font and characters size and use appropriate colors easy to be observed by the reader.

2. Please use the same writing for Ca2+ through the entire manuscript.

3. In Figure 2, please flip vertically the following words: Wnt-1 agonist, NO-NSAIDs and GSK-3β antagonists, for easier layout of the figure.

4. Paragraphs, in some cases, would need to be sub-divided to simplify the reading and maybe converge in vitro and in vivo work separately.

5. Title of each paragraph is often too long, losing the impactful meaning of it.

6. English needs editing, many typos are present in the text and please use appropriate wording.

Author Response

I thank the Reviewer for his/her comments and as requested, more work was done on the Ms to improve it for clarity, as suggested.

1. The quality/resolution of all the figures needs to be increased, adjusting also the font and characters size and use appropriate colors easy to be observed by the reader.

1. Quality and Fig resolution have been adjusted accordingly.

2. Please use the same writing for Ca2+ through the entire manuscript.

2. Corrected.

3. In Figure 2, please flip vertically the following words: Wnt-1 agonist, NO-NSAIDs and GSK3β antagonists, for easier layout of the figure.

3.Corrected

4. Paragraphs, in some cases, would need to be sub-divided to simplify the reading and maybe converge in vitro and in vivo work separately.

4. Paragraphs have been subdivided accordingly.

5. Title of each paragraph is often too long, losing the impactful meaning of it.

5.Corrected

6. English needs editing, many typos are present in the text and please use appropriate wording.

6. Ms editing and the use of appropriate wording provided

Reviewer 3 Report

Overall Marchetti has provided a rather thorough overview of the potential contributions of Wnt/B-catenin signaling in PD. There are, however, several places the review could be improved.

There are several places where the information provided is either incomplete or not factually accurate

The SNpc projects to both the caudate and the putamen not just the caudate (p 1).

Neurotoxin and toxicants are not interchangeable terms. Toxins are naturally occurring and toxicants are manmade -MPTP is confusingly referred to as both without sufficient explanation.

MPTP is a protoxin, not a toxin. Its metabolite MPP+ is the toxin- this distinction isn't clear on p 7-8.

The use of the first person narrative "I" is not appropriate for a published review article.

There are several places in the text where incorrect grammar and vocabulary are used - e.g. "evidences" is not a word, "Galli and co" - colleagues or " et al", "opposedly" is not a word, singular nouns with plural verbs, etc.

The molecular labels on the figures are illegible and must be in much higher quality resolution.

Author Response

There are several places where the information provided is either incomplete or not factually accurate

I thank the Reviewer for his/her important comment. A more accurate information has been given , as requested.

The SNpc projects to both the caudate and the putamen not just the caudate (p 1).

corrected accordingly (introction)

Neurotoxin and toxicants are not interchangeable terms. Toxins are naturally occurring and toxicants are manmade -MPTP is confusingly referred to as both without sufficient explanation.

Accordingly section 3 has been expanded (see section 3)

MPTP is a protoxin, not a toxin. Its metabolite MPP+ is the toxin- this distinction isn't clear on p 7-8.

I thank the Reviewer for his/her important indications and provided more accurate information on the subject (see section 3)

The use of the first person narrative "I" is not appropriate for a published review article.

The use the first person has been corrected accordingly

There are several places in the text where incorrect grammar and vocabulary are used - e.g. "evidences" is not a word, "Galli and co" - colleagues or " et al", "opposedly" is not a word, singular nouns with plural verbs, etc.

Incorrect grammar/vocubulary ecc. have been corrected

The molecular labels on the figures are illegible and must be in much higher quality resolution.

Molecular labels and higher resolution images have been provided.